# Cities: Complexity, theory and history

**Scott G. Ortman**[1,2]*, **José Lobo**[3], **Michael E. Smith**[4]

**1** Department of Anthropology, University of Colorado Boulder, Boulder, Colorado, United States of America, **2** Santa Fe Institute, Santa Fe, New Mexico, United States of America, **3** School of Sustainability, Arizona State University, Tempe, Arizona, United States of America, **4** School of Human Evolution and Social Change, Arizona State University, Tempe, Arizona, United States of America

* scott.ortman@colorado.edu

**Data Availability Statement:** The data underlying the results presented in the study are available at https://core.tdar.org/project/392021/social-reactors-project-datasets.

**Funding:** Portions of this research were supported by a grant from the James S. McDonnell

## Abstract

In recent decades researchers in a variety of disciplines have developed a new "urban science," the central goal of which is to build general theory regarding the social processes underlying urbanization. Much work in urban science is animated by the notion that cities are complex systems. What does it mean to make this claim? Here we adopt the view that complex systems entail both variation and structure, and that their properties vary with system size and with respect to where and how they are measured. Given this, a general framework regarding the social processes behind urbanization needs to account for empirical regularities that are common to both contemporary cities and past settlements known through archaeology and history. Only by adopting an explicitly historical perspective can such fundamental structure be revealed. The identification of shared properties in past and present systems has been facilitated by research traditions that define cities (and settlements more broadly) as networks of social interaction embedded in physical space. *Settlement Scaling Theory* (SST) builds from these insights to generate predictions regarding how measurable properties of cities and settlements are related to their population size. Here, we focus on relationships between population and area across past settlement systems and present-day world cities. We show that both patterns and variations in these measures are explicable in terms of SST, and that the framework identifies baseline infrastructural area as an important system-level property of urban systems that warrants further study. We also show that predictive theory is helpful even in cases where the data do not conform to model predictions.

## Introduction

In recent decades technological and social developments have stimulated the collection of unprecedented amounts of information concerning what individuals, business and organizations do in cities. For some this "big data revolution," driven by satellite imagery, global positioning systems, smart phones, digital cameras, remote sensing and social media, provides opportunities for engineering "smart" cities that manage people and resources more efficiently and effectively. For others, the availability of more data, and different types of data, has made it possible to investigate cities in novel and multidisciplinary ways. The development of new

Foundation (#220020438, to SO): https://www.
jsmf.org/. Additional support has been provided by
the Arizona State University / Santa Fe Institute
Center for Biosocial Complex Systems. There was
no additional external funding received for this
study.

**Competing interests:** The authors have declared
that no competing interests exist.

pathways for studying cities has coincided with a realization that many of the challenges facing humanity also involve urban areas. This has encouraged researchers to treat the city as the unit of analysis, as the Nobel-Prize winning economist Paul Romer [1] has argued:

"The urban environment that humans are so busily creating is many things: a biological environment, a social environment, a built environment, a market environment, a business environment, and a political environment. It includes not only the versions of these environments that exist inside a single city, but also those that are emerging from the interaction between cities. Our understanding of the urban environment will draw on existing academic disciplines, but it will also develop its own abstractions and insights."

These various trends are coming together in a new approach to cities, *urban science*, which draws on existing research traditions and recent developments in urban economics, economic geography, labor economics, urban sociology, urban ecology, spatial data analysis and network science [2–4]. The scope and purpose of urban science has been articulated well by Michael Batty [5: 998]:

"Urban science deals with the structure and functioning of cities, and the generic laws that seem to govern cities everywhere insofar as they can be articulated. . . a science of human behavior as it applies to cities. . . This is not the science of the physics of buildings or energy flows in cities (although it clearly relates in part to some of these aspects), it is the science of people flows, flows of goods, and the flow of information and ideas and the extent to which all these can be generalized over city size and scale."

A prominent aspect of urban science is the notion that cities are complex systems [4, 6–13]. This view can be traced to Jane Jacobs' discussion of cities as "problems of organized complexity" which to her meant "dealing simultaneously with a sizeable number of factors which are interrelated into an organic whole" [14: 432]. What does it mean to call a city a complex system? We believe a good place to begin is with general properties of such systems. First, they are *non-extensive*, meaning that their overall properties are emergent, and are not simple summations of their parts. As was famously stated by Phil Anderson [15], "more is different." Second, they are *non-intensive*, in the sense that the properties of their parts covary with the total system size. When one divides a complex system into parts (such as people or locations in a city) the properties of each part are not only dissimilar, but they also depend on the size of the system of which they are parts [16]. These considerations suggest a city only makes sense as a complex system when all its constituent parts come together as an interacting whole, which is to say, when the entire city is the object of study.

Goldenfeld and Kadanoff [17: 87] suggest further that a hallmark of complexity is structure with variations, where the structure results from interactions among a system's components. This directs us to consider the connections between structure and variation: to identify systemic features and generative processes shared by cities in different settings *and* to specify how meaningful variations (as distinguished from noise) emerge with respect to these generative processes. To truly study cities as complex systems, then, one needs to examine settlements across geographies and eras to identify commonalities, observe variations, and determine how the two are related. This requires a historical perspective, engaging not only with contemporary systems but also with historical and archaeological records.

Some in the urban science research community hold the view that because cities are complex systems, they are resistant to theories that generate testable predictions. This view is fueled in part by a conception of cities as super-organisms, and a misconception that the properties

of such super-organisms (e.g. ant colonies) are inherently chaotic and unpredictable [18]. While it is true that the detailed evolution and properties of every city are unique, and the properties of cites are both non-extensive and non-intensive, researchers have developed predictive theory for many other complex systems, ranging from stars [19] to organisms [20]. Also, the fact that empirical regularities regarding the aggregate properties of cities and urban systems can and have been established by geographers [e.g., 21] suggests that statistical predictions regarding their properties can be constrained by general considerations, which in turn allow the specifics of individual cities to be assessed quantitatively.

So, the fact that each city is unique when viewed in detail does not imply that predictive theory of cities is impossible. It merely indicates that such theory needs to deal with cities and urban systems as integrated wholes. The approach must focus on properties that can be captured by common measurements related to the functional role of cities in human societies; and potential explanations must likewise invoke social processes that are common to cities across space and time. The approach cannot simply involve projecting contemporary social and economic arrangements into the past, as this would break down as soon as contemporary technologies and institutions ceased to apply. Instead, a productive approach should generate specific predictions that can be tested using data from any settlement system. In the same way that the theory of biological evolution applies to both the fossil record and contemporary life, a good theory of urbanism should apply to both the archaeological record and contemporary urban systems [22, 23].

In this paper we apply this approach to studying cities as complex systems. We conduct a broad empirical investigation of two common measurements—population and area—that are widely-available for both contemporary and ancient systems; and we evaluate patterns and variations in their relationship with respect to a framework known as *settlement scaling theory* (SST). In this approach the essential feature of human agglomerations is that they consist of social networks embedded in physical space. Since these can be of any scale, boundaries between traditional categories of city, town and village are somewhat illusory, and the terms *city* and *settlement* can be used interchangeably. The contemporary data are from the OECD's Cities of the World database, which provides estimates the area and population of functional urban areas for every nation in 2015; and the historical and archaeological data derive from completed and in-progress studies by the Social Reactors Project (https://www.colorado.edu/socialreactors/) using archaeological and historical settlement pattern data for eleven different regions across ancient North America, Mesoamerica, the Andean region, Classical Antiquity, Medieval Europe, and Early China.

Our exploration leads to the following points. First, relationships between population and area vary across settlements and urban systems, both in the contemporary world and in the past. Second, for those systems where the units of observation capture spatially embedded social networks, there are strong regularities in the relationship between population and area that correspond closely to predictions of SST. The fact that such statistical regularities, which share specific and predictable values, occur across such a wide range of societies makes a strong case that the fundamental social process behind the emergence of cities everywhere is the concentration of social interaction in space. Finally, we find substantial variation in the baseline areas of both past and contemporary settlement systems, but this variation is not haphazard. We find that the least developed regions of the world today have baseline areas that approach those of preindustrial civilizations, that there are a number of reasons to view baseline area as a simple index of socio-economic development, and that scaling intercepts (baseline areas) and slopes (exponents) are closely related in functional urban areas (FUAs) as defined by the OECD. These results suggest it would be profitable to examine the relationship between baseline area and living standards more closely, and to seek data for testing SST

predictions regarding the determinants of baseline area. We hope this investigation will convince readers: 1) that a predictive theory of cities as complex systems is possible; and 2) that a broad historical and comparative perspective is essential for the ultimate success of this effort.

## Defining the city

Despite being separated by thousands of years of cultural, social and technological development, ancient and contemporary population centers have enough in common that the term "city" can be meaningfully applied to both [24–26]. But what is meant by this term? Lewis Mumford [26: 29] described the city as "a geographic plexus, an economic organization, an institutional process, a theater of social action, and an aesthetic symbol of collective unity." Louis Wirth [27] proposed that a city is a permanent settlement of heterogeneous and interdependent individuals living and working at high population densities, and Richard Sennett [28: 39] suggests that "a city is a human settlement where strangers are likely to meet." Architectural historian Spiro Kostof [29: 37] observes that "cities are places where a certain energized crowding of people takes place," and the urban economist Edward Glaeser [2: 6] describes cities as "the absence of physical space between people and companies. They are proximity, density, closeness." For Batty and Ferguson [30: 753] cities are dense agglomerations where people come together to trade and engage in diverse social relationships and the friction of distance is minimized. A recent urban economics textbook [31] defines a city as a geographical area with a higher concentration of individuals and activities than the surrounding area. Finally, O'Flaherty [32] gives a minimalist definition: cities are circumscribed and relatively small areas where concentrated individuals perform almost all the activities of their lives (sleeping, working, eating, playing).

A common thread in these various characterizations is the view that the essence of urbanism is not physical space per se, but frequent and intense social interactions among a diversity of individuals and organizations within a space [33]. Two key assumptions help one translate this common thread into a functional definition of cities. First, movement entails costs: social interactions in space have, throughout history, involved travel, which carries energetic, opportunity, and monetary costs [34]. Second, human effort is bounded—for any given transportation technology, humans can only move so fast per unit time and spend only so much time in transit [35]. Together, these assumptions support the definition of functional urban areas (see below) as spatial units whose outlines encapsulate daily flows of people, goods, and information [36]. How much distance can be covered in a day, and at what cost, is strongly dictated by available technologies and infrastructure and their local implementations. One can thus conceive of a city as a self-organized social and spatial arrangement where interactions among people are only impeded by transaction, transportation, opportunity, and matching costs (the cost of matching any one individual's resources, needs, interests and skills with those of another individual [37]). The social networks comprising cities are "well-mixed" in the sense that all types of interactions among individuals are possible, even if only a portion of the possible interactions occur per unit time.

## Defining urban boundaries

The spatially embedded networks of interaction discussed above need not have firm boundaries, but because it is not yet feasible to directly map all the physical encounters that comprise such networks one needs to define areas that contain most of these interactions if one wishes to measure the populations or other properties of cities. Thus, the most fundamental issue in the study of cities as complex systems is defining urban boundaries.

It turns out to be much easier to define such boundaries in an archaeological context than it often is in contemporary contexts. Despite the many challenges associated with archaeological evidence, one of its advantages is the intrinsic correspondence between settlement boundaries and spatial patterns of daily social interaction. In ancient societies, people walked, or, in rare cases, rode on animals or in carts, along paths that were much more uneven than modern roads. Most people who worked within settlements lived close to—or even at—their place of work. Individuals rarely lived in one settlement and worked in another [38, 39]. In addition, most workers were farmers who regularly walked to fields that lay outside the settlement [40]. So, for such settlements if there were "commuter flows" at all they involved farmers commuting between settlements and their fields. Commuting served to disperse people for individual farm work, with most social interactions being confined to the settlement area itself. As a result, in the preindustrial world the physical settlement (based on the circumscribing or built-up area) and the functional settlement (based on social mixing patterns) were essentially one and the same [22]. As a result, when archaeological remains are visible on the modern ground surface it is relatively straightforward to define settlement areas as contiguous areas of such remains, with settlement boundaries being drawn where discarded artifact densities decline sharply, or where there is a substantial gap in the distribution of remains [41].

The situation is very different in the contemporary world, where fast and inexpensive transport and national political integration erodes correspondences between the administrative, physical, and social aspects of cities. There is a recognition that the relevant spatial units are areas within which most residents travel and interact on a daily basis. But defining such units in a consistent way across levels of socio-economic development remains a challenge. The U.S. Census Bureau has a long-standing, and arguably the most consistent, definition of functional cities, known as the Metropolitan Statistical Area (MSA), dating back to the 1950s and updated annually (For a history of the Metropolitan Statistical Area (MSA) concept go to https://www.census.gov/history/www/programs/geography/metropolitan_areas.html). An MSA consists of a core county or counties within which lies an incorporated city (a politico-administrative entity) with a population of at least 50,000 people, plus adjacent counties having a high degree of social and economic integration with the core counties as measured through commuting ties. An MSA is in effect a unified labor market within which goods, labor and information flow on a daily basis, and this in turn implies the existence of a spatially embedded network of intense socio-economic interaction [42].

Because of its unique socio-economic relevance, the concept of the metropolitan area has been adopted by other major national and international statistical offices, including the OECD that generated the data we examine here. The OECD version of a functional city, known as a Functional Urban Area or FUA, is defined in several steps. First, global human settlement is modeled by overlaying a 1-km grid over a satellite image of the earth's surface and estimating the fraction of each cell that is built up; and then this layer is integrated with national census data to allocate population into the same 1-km grid. Second, an *urban center* is defined as a contiguous region of 1-km grid squares containing at least 1,500 residents per cell, or at least 50% built-up surface share per cell, and a minimum population of 50,000, and the edges are smoothed [43]. Third, a *city* is defined as a contiguous set of local units (administrative or statistical) with at least 50% of their populations within an urban center. Finally, a *commuting zone* is defined by adding to the city a) other cities where at least 15% of employed persons work in the target city; b) local units where at least 15% of employed residents work in the target city; c) local units with less than 15% commuting that are nevertheless surrounded by a city's commuting zone; and d) dropping local units with more than 15% commuting that are discontinuous with the rest of the commuting zone. Functional urban areas are then the summed area and population of the city and its associated commuting zone [44].

These spatial units are not perfect in that they rely on the identification of built areas from satellite imagery at a resolution of 1-km, and the intersection of these areas with the finest spatially explicit local units of population enumeration in each nation. One might expect the algorithm discussed above to capture areas with different mixes of built, urban, and rural uses across nations. Nevertheless, the OECD method defines and measures functional urban areas in a consistent way and has the advantage of having been applied across the globe, making it feasible to compare relationships between population and area on a global scale. As such, it represents an important resource for a broad investigation of patterns and variations in urban systems.

## A model for agglomeration effects

Insights from anthropology, economics, sociology, and complexity science provide a foundation for viewing cities as systems that emerge from the interplay of centripetal and centrifugal forces; specifically, the socio-economic advantages of concentrating human populations in space vs. the associated costs of doing so. These are known as "agglomeration effects", and they constitute foundational concepts for explaining the emergence and persistence of cities everywhere [45–47]. Urban agglomeration effects reflect the systematic changes in average socio-economic performance, land-use patterns, and infrastructural characteristics of cities as functions of their size. Settlement scaling theory seeks to explain the emergent properties of cities in terms of these processes.

The foundational assumptions of SST are: (a) human interactions are exchanges of material goods, services and information that take place in physical space; (b) the intensity, productivity and quality of individual-level efforts are mediated and enhanced through interaction with others (social networks); (c) any human activity can be thought of as generating benefits and incurring costs (especially the costs of moving people and things in physical space); (d) human effort is bounded; and (e) the size (scale) of a human agglomeration is both a consequence and a determinant of the agglomeration's productivity. These assumptions provide the microfoundations for predicting aggregate scaling phenomena in terms of the behavior of representative agents and their (economic and non-economic) interactions [48]. Note especially that capitalist markets or specific types of political organization are not ingredients of the theory.

In this approach, spatially concentrated social networks and associated costs and benefits, which can take different institutional and cultural forms, are sufficient to predict the scale-dependent properties of cities as complex systems. As a result, population size and density (by way of areal extent) play a central role in SST, both as important characteristics to be accounted for and as drivers of other important characteristics of settlements. In this way, SST builds from previous research traditions that have assigned an important explanatory role to scale (size) [15, 49]. Although there are other important attributes of urban form besides population size and density [see, e.g., 50], the settlement scaling framework provides an argument grounded in first principles for a privileged role for population size and density.

The settlement scaling framework has been presented in detail elsewhere [51–55] so here we limit our discussion to the relationship between population and area that is the focus of this paper. We begin by proposing that when individuals arrange themselves socially in physical space, they do so in a way that balances the benefits of interacting with others with the costs of moving around to do so. When settlements are small and unstructured, the cost of such movement is given by $c = \epsilon L$, where $\epsilon$ is the energetic cost of movement and $L$ is the transverse distance across the area over which people have settled. In this circumstance the distance is proportional to the square root of the circumscribed area containing the settlement, $L \sim A^{1/2}$. The social benefits resulting from such movement are simply the number of interactions a

person has per unit time, given by $i = a_0 l N / A$, where $l$ is the average length of the path traveled by an individual in a day, $a_0$ is the distance at which interaction occurs, and $N/A$ is the average population density within the circumscribing area. Different individuals will experience different interactions, of course, but here the average over the population is sufficient. These interactions are mostly intentional so that they can be translated into net benefits, $y$, by considering that there is some average net energetic consequence of an interaction, across all types of interactions that can occur $\hat{g}$, such that $y = \hat{g} a_0 l N / A$. Then, by assuming that costs and benefits balance, $c = y$, there is a spatial equilibrium of costs and benefits, $\epsilon A^{1/2} = \hat{g} a_0 l N / A$, and this simplifies to:

$$A(N) = aN^{2/3}, \qquad (1)$$

where the coefficient or *pre-factor* $a = (\hat{g} a_0 l / \epsilon)^{2/3}$. One can think of $a$ as the net attractive "force" (resources per unit time per unit area = power density) an individual exerts on others through his/her interactions.

Eq (1) proposes that as the number of people who mix socially on a regular basis increases, the total area taken up by these people will grow more slowly than the number of people, such that the area taken up by each person will decrease. Notice, however, that in order to see this process empirically one must be able to define the circumscribing area $A$ over which the social mixing of $N$ people occurs on a regular basis, and indeed, a circumscribing area needs to be a reasonable way of characterizing the area over which people are distributed. The pre-factor $a$ in Eq (1) varies in accordance with the strength of social interaction and transportation costs and can change over time with changes in transport and social institutions. It is also generally assumed that unit movement costs are independent of scale, but as we shall see, this may not be the case for rapidly developing nations today. Eq (1) also expresses the way in which basic properties of settlements, especially their area and density, are *non-intensive*, meaning that they vary with the system (population) size. Below, we show that these properties also vary depending on the choice of spatial unit for measurement. Both are hallmarks of complex systems.

Eq (1) applies to small and spatially unstructured settlements, but as settlements grow the inhabitants must increasingly set aside some of the land area, $A_n$, for roads, paths, public spaces and public infrastructure so that residents can continue to move around and mix socially. This is the actual area over which the spatial equilibrium of interaction costs and benefits occurs, and as a result it is necessary to specify the relationship between people and this "network" area, and also to actually measure the network area. We assume that on average the distance $b$ between people is set in accordance with the current population density, such that $b \sim (A/N)^{1/2}$. This can be justified by the observation that historically infrastructure has been built or expanded in urban areas *mainly* in response to expansion of the resident population [56–59]. Thus, one can think of $b$ as the length and width of street-frontage per resident in a city. Under this model, the total area of the access network is:

$$A_n = Nb = A^{1/2}N^{1/2}. \qquad (2)$$

From here, one can substitute $aN^{2/3}$ for $A$ and simplify, leading to:

$$A_n = a^{1/2}N^{5/6}. \qquad (3)$$

Eq (3) implies that, as settlements in a given context grow, movement and interaction become increasingly structured by the access network and its associated public spaces, and that the area of this network grows with population more rapidly than the circumscribing area, leading the exponent of the population-area relationship to transition from 2/3 to 5/6. There is

still an economy of scale in space use per capita, but the exponent of the growth rate of the built area with population is slightly higher than it is with respect to a circumscribing area.

It is important to emphasize that Eqs 1 and 3 are mean-field models that predict the average rate of increase in area, relative to population, as specified by the exponent of $N$. Another way of saying this is that they predict expectation values for the area of a settlement, given its population and the baseline area per person in that system, reflected in the pre-factor $a$. In these models all parameters besides $A$, $N$ and $b$ are scale-invariant quantities, but they need not be constant across systems or over time. In addition, there are a range of additional factors one would expect to be involved in determining the population and area of any specific settlement. Given these additional considerations, a more exact way of writing Eq 3 is:

$$A_{ni}(N_i, t) = a_0(t)N_i(t)^{5/6} * e^{\varepsilon_i(t)}, \qquad (4)$$

where $a_0(t)$ is standing in for $a^{1/2}$. This notation indicates that the pre-factor of the scaling relationship is specific to a particular system at a particular time, and $e^{\varepsilon_i(t)}$ captures the range of influences unique to each city that lead to a deviation of infrastructural area in any given settlement from the average expectation. We represent this deviation as an exponential so that it will take the form of a Gaussian random variable following natural log transformation. To see this, one can take the natural logarithm of Eq 4:

$$\ln[A_{ni}(N_i, t)] = \ln[a_0(t)] + (5/6) * \ln[N_i(t)] + \varepsilon_i(t), \qquad (5)$$

and then express Eq (5) as the ensemble average. Since by definition $\langle \varepsilon_i(t) \rangle = 0$, it can be dropped from the ensemble average, leaving the following result:

$$\langle \ln[A_{ni}(N_i, t)] \rangle = \ln[a_0(t)] + (5/6) * \langle \ln[N_i(t)] \rangle. \qquad (6)$$

This linear function is an exact expression that relates the mean of the log of infrastructural area across cities in an urban system at time $t$ to the mean of the log of population across those same cities at that time. Eq 6 makes clear what can and cannot be predicted using SST models. One cannot exactly predict the area of any given city based on its population, or vise-versa, but one can predict the average relationship and the relationship of the averages. In addition, the average of the deviations from the expectation value across cities should sum to zero, which is to say, the deviations should follow a standard normal distribution in log-transformed variables (and thus a log-normal distribution in the original variables). Finally, there is a concrete prediction regarding the numerical value of the coefficient that relates population to area for the log-transformed variables (and thus the exponent of the relationship for the original variables), but the numerical value of the intercept for the log-transformed variables (and the pre-factor for the original variables) is a system-specific and time-specific property that is independent of population and area. In the analyses that follow we focus on these predictions with respect to a range of past and contemporary systems.

The relationships between settlement population and area discussed above presuppose the ability to define areas over which daily interactions took place. For small and amorphous settlements, this is the area circumscribing the interacting population; and for larger cities, it is the area of residences, workplaces, shops, and transport infrastructure within which daily social mixing occurs. Given this, a key question is the extent to which the spatial units defined by urban scientists correspond to the networks of interaction in space envisioned in these models. This question is also addressed in the discussion of our results below.

## Results

We assess relationships between the population and area of settlements, based on archaeological and OECD methods for defining settlement boundaries, through ordinary least-squares regression of the log-transformed measures of population and area for these spatial units. As shown above, this is feasible because $X(N) = X_0 N^\beta$ and $\log[X(N)] = \beta\log[N]+\log[X_0]$ are equivalent. As a result, the slope of the best-fit line for the log-transformed data ($\beta$) is also an estimate of the exponent of the scaling relation, and the y-intercept of the best-fit line ($\log[X_0]$) is an estimate of the logarithm of the pre-factor ($X_0$) of the scaling relation. Because population and area reflect the same quantities across all systems the results are in principle directly comparable, to the extent that the spatial units defined for each system capture the same social units. What we will see, however, is that it is difficult to capture actual network areas across contexts using a single definition of spatial units.

### Archaeological and historical data

Table 1 summarizes the available data and estimated population-area relationships for a variety of archaeological and historical systems. Collectively, these systems span four continents and 7,000 years of history. The raw data were collected in the field or in archives by regional experts, and each case is discussed individually in the study cited in the table, but in all cases scaling relations have been recalculated for this study using ten persons as a minimal population size for inclusion. The archaeological cases clearly represent time-averaged data for settlements dating to different time periods. In some cases, population and area estimates are available for the same settlement during different periods, but in most cases the measurements reflect the moment when a settlement reached its maximal extent. Importantly, populations

**Table 1. Population-area relationships in archaeological and historical systems.**

| Label | Region | Date Range | Sample Size | Population Range | Slope | SE | Intercept | SE | $R^2$ | Reference |
|---|---|---|---|---|---|---|---|---|---|---|
| Basin of Mexico Dispersed | Northern Mesoamerica | 1150 BCE-1520 CE | 1147 | 10–2,000 | 1.02 | .011 | -2.78 | .044 | .88 | [55] |
| Basin of Mexico Towns | Northern Mesoamerica | 1150 BCE-1520 CE | 1653 | 10–212,500 | .81 | .011 | -2.19 | .052 | .75 | [55] |
| Chifeng, PRC | Northern China | 6000 BC-1100 CE | 342 | 10–18,240 | .64 | .020 | -1.88 | .100 | .75 | (Ortman and Cooper, in prep.) |
| Classic Maya | Southern Mesoamerica | 600–850 CE | 501 | 10–8,650 | 1.47 | .041 | -6.35 | .143 | .72 | [64] |
| Izapa, Mexico | Southern Mesoamerica | 700–100 BCE | 39 | 15–445 | 1.41 | .094 | -4.16 | .348 | .86 | [64] |
| Lower Santa Valley, Peru | Coastal Peru | 1000 BCE-1532 CE | 84 | 10–6,470 | .65 | .048 | -2.17 | .199 | .69 | [65] |
| Mantaro Valley, Peru | Highland Peru | 1000–1532 CE | 96 | 15–23,750 | .63 | .062 | -2.41 | .331 | .53 | [66] |
| Medieval Europe | Western Europe | ca. 1300 CE | 173 | 1,000–200,000 | .71 | .026 | -2.13 | .247 | .81 | [67] |
| Mesa Verde Region, USA | US Southwest | 1060–1280 CE | 278 | 15–960 | .66 | .076 | -2.61 | .262 | .22 | [68] |
| Middle Missouri, USA | US Great Plains | 1200–1886 CE | 35 | 35–875 | .64 | .081 | -2.35 | .438 | .65 | [68] |
| Mixteca Alta, Mexico | Southern Mesoamerica | 1400 BCE—1520 CE | 1174 | 10–31,995 | .83 | .011 | -2.20 | .052 | .84 | [69] |
| Roman Empire | Mediterranean Basin | 100 BCE—300 CE | 53 | 700–1,000,000 | .65 | .034 | -1.93 | .321 | .88 | [70] |

Note: All regressions are significant at the P < .0001 level.

were estimated using different proxies and methods that are appropriate for each case, ranging from historical records (Medieval) to house densities in sample areas (Roman), direct house counts across settlements (Mantaro, Lower Santa, Mesa Verde, Middle Missouri, Maya and Izapa), and artifact densities within settlement areas (Chifeng, Basin of Mexico, Mixteca Alta). In all cases households were converted to individuals by multiplying the estimated house count by five. Fig 1 presents the scatter plots and fit lines, and quantile plots of the distribution of residuals, for the data from each system. Fig 1 demonstrates that, despite the vagaries of archaeological evidence and variation in methods used to estimate population, the relationship between log-transformed population and area estimates is generally well-described by a linear function, and in most cases the residuals to the fit line closely follow a standard normal distribution. Unfortunately, with archaeological data, it is rarely possible to distinguish measurement error from true fluctuations, and for some systems area is involved in constructing the population estimates. As a result, residuals for specific settlements and the $r^2$ values in Table 1 have limited interpretive value.

Fig 2 provides a visual summary of the regression results for these systems, with cases listed in order of their respective scaling parameters. Fig 2A shows the point estimates and confidence intervals for the scaling exponent $\beta$, and shows the predicted range based on SST using dashed lines. The results show that about half of the archaeological cases (Mantaro, Chifeng, Middle Missouri, Lower Santa, and Mesa Verde) exhibit scaling exponents whose confidence intervals easily overlap with the value predicted by the amorphous settlement model, where $\beta$ = 2/3. These systems, except for the Roman Empire, consist of settlements defined as elliptical shapes that enclose populations ranging from tens to no more than a few tens of thousands of people. Transport infrastructure can occur in the spaces between houses, but there is little sense that the positions of houses were determined by the transport infrastructure. Thus, the spatial units for these systems correspond well to the image of amorphous settlements, and their scaling exponents follow suit.

A few other systems exhibit scaling exponents between 2/3 and 5/6. These are larger-scale systems in which housing in the largest settlements is arranged with respect to a street system (Medieval, Basin of Mexico Towns) or other economic infrastructure (lama-bordo terracing in Mixteca Alta). The settlements in these systems correspond more closely to the image of networked settlements, and their exponents reflect this as well. In this context, the exponent for the Roman Empire seems lower than expected in that it corresponds more closely to the expectation of the amorphous settlement model than the networked settlement model, despite the fact that this system is the largest-scale archaeological system in the analysis, and many Roman cities were organized around street networks [60]. The reasons for this are unclear, but simple explanations are possible. For example, the areas of Roman cities are also often estimated using simple shapes due to the fact that many lie beneath still-inhabited cities. In addition, the population estimates for these settlements assume the number of residents per residential unit was scale independent, but one could imagine this number increasing slightly in larger and denser cities, and this would serve to decrease the scaling exponent. It is also possible that movement in Roman cities was structured in time to a greater extent than is assumed by the networked settlement model. For example, vehicle-based transport may have been limited to certain times of the day, as occurs in some European historic city centers today. This issue deserves additional study.

Finally, a few of these systems (Basin of Mexico dispersed settlements, Maya and Izapa) exhibit scaling exponents that are superlinear, meaning that larger settlements are less dense than smaller ones. These cases clearly do not conform to SST models. In these systems, settlement boundaries defined by archaeologists do not correspond to the areas over which daily social mixing took place. The reason appears to be that in these cases "settlements," if they can

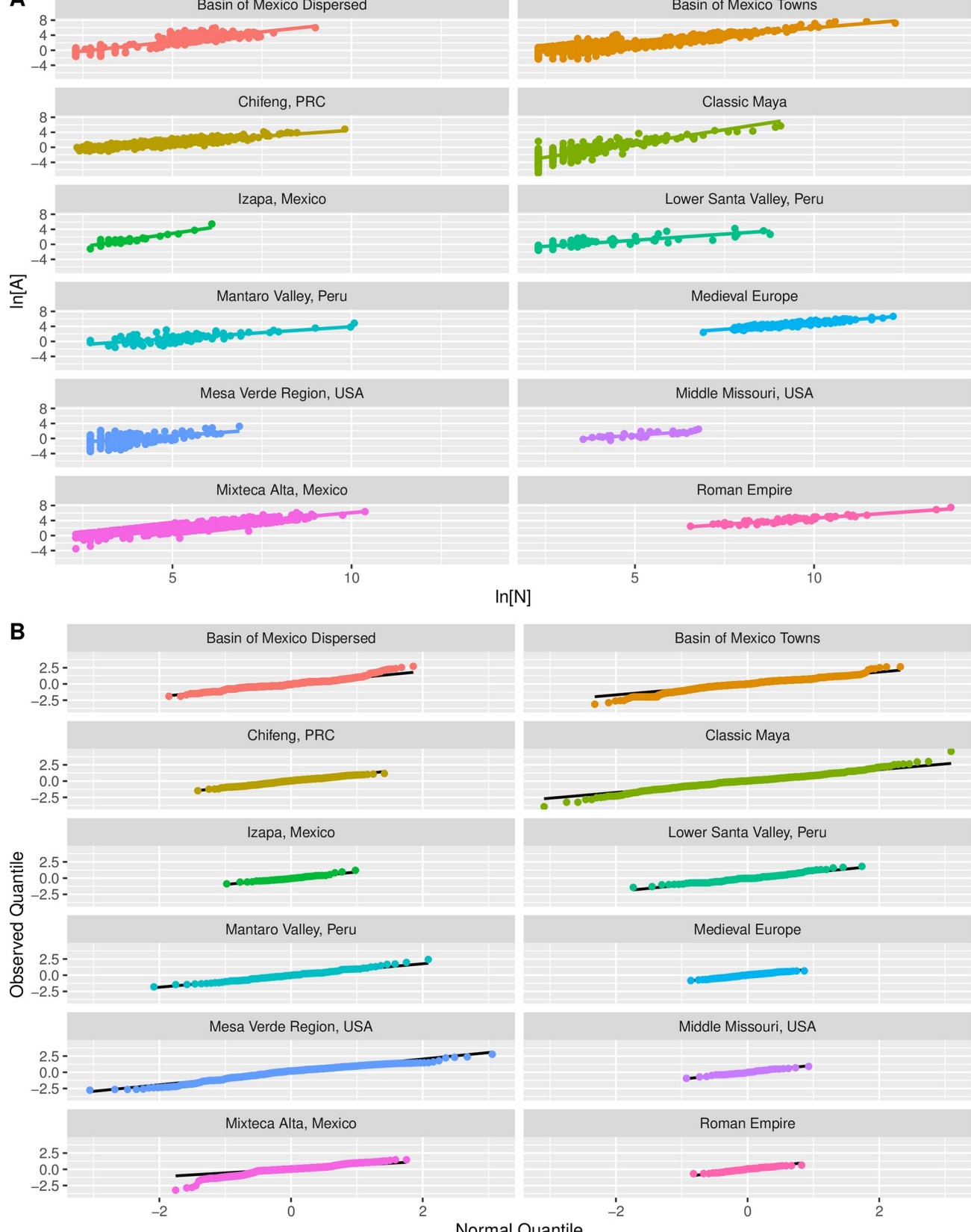

**Fig 1.** Population-area relationships for archaeological and historical systems: A) regression plots; B) quantile plots of residuals.

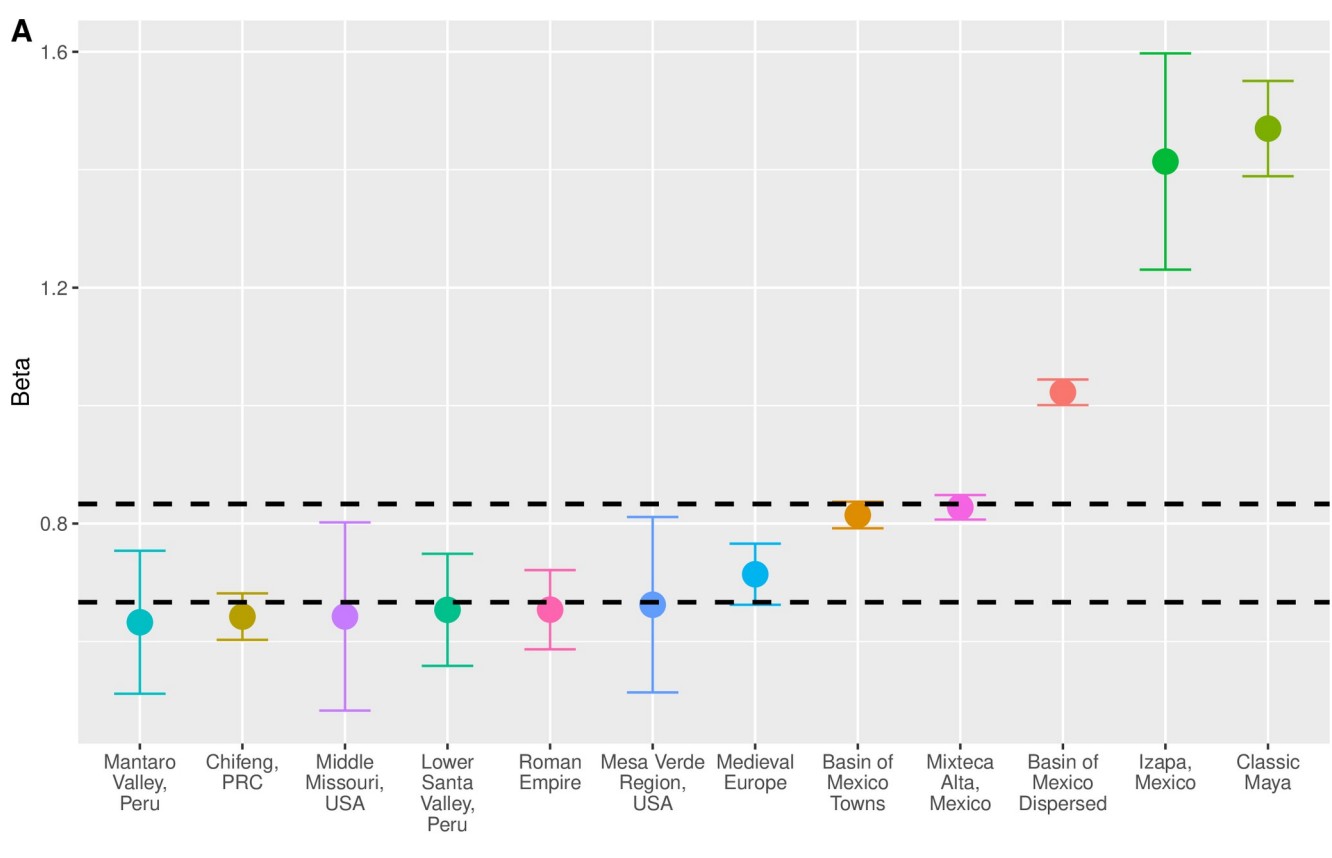

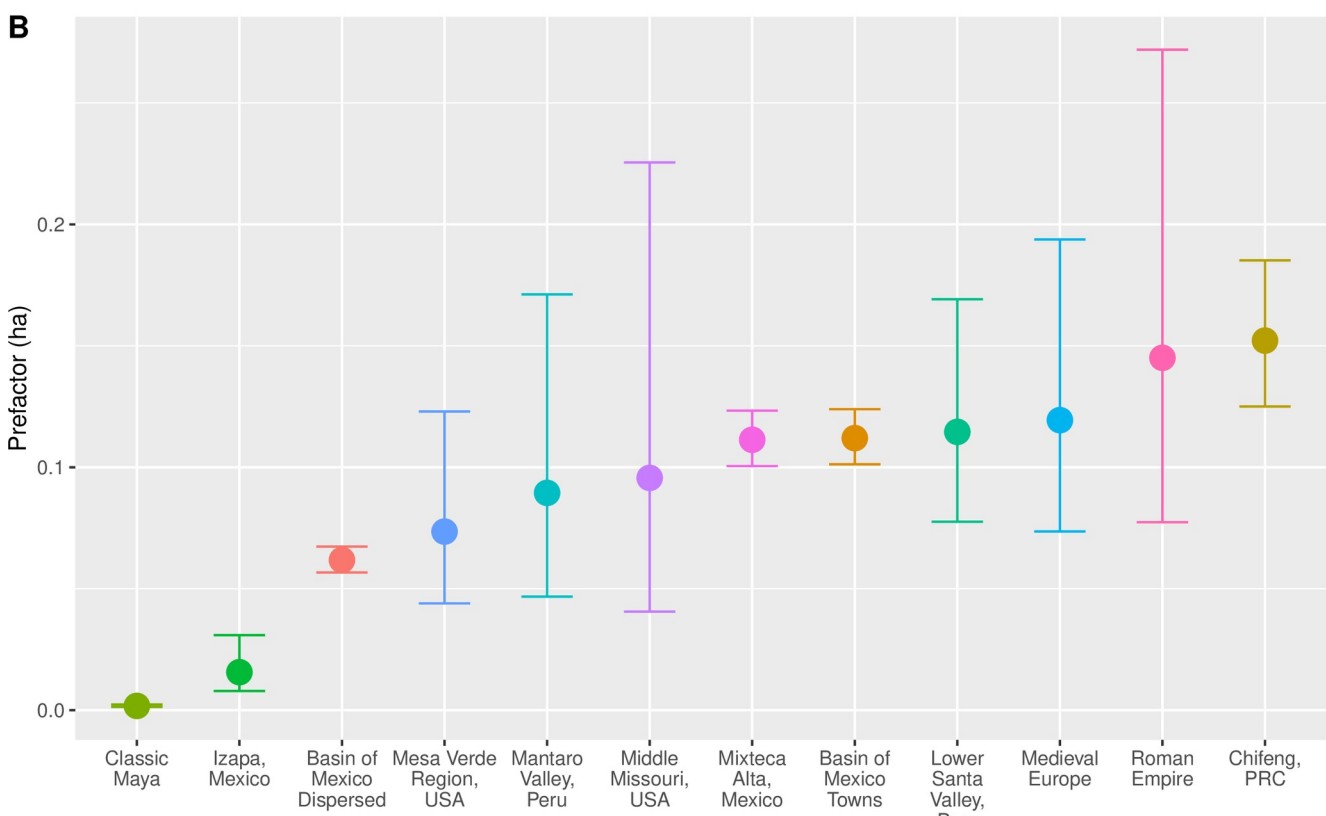

**Fig 2.** Scaling relations for archaeological and historical systems: A) Exponents; B) pre-factors.

still be referred to as such, consist of residences interspersed with agricultural land, centered on a smaller civic area. As a result, the area enclosed within a settlement boundary expands much faster than the area over which the people inside actually interacted socially. It is also possible that in these systems patterns of social interaction were qualitatively different than is the case in of systems that exhibit densification with scale, such that social mixing over space may not be a good characterization.

Fig 2B summarizes point estimates and confidence intervals for the pre-factors of the population-area relationship across archaeological and historical cases, with the estimates arranged in increasing order. This parameter represents the baseline area utilized by an individual in the smallest settlements. The confidence intervals are large and overlap in many cases, so at least some of the observed variation is due to measurement and sampling error. It is also important to mention that in several cases (Basin of Mexico, Middle Missouri, Mantaro, Chifeng) time series analyses have not been able to identify statistically significant variation in either exponents or pre-factors across archaeological periods. This suggests that in many cases scaling relationships were either stable or fluctuated around long-term averages in preindustrial societies. Nevertheless, there is some meaningful geographical variation in scaling pre-factors reflected in Fig 2A, with baseline areas in the Chifeng and Imperial Roman systems being larger than that of several other systems, and pre-factors for the three de-densifying systems (Maya, Izapa and Basin of Mexico dispersed) being notably smaller. There also seems to be a general correspondence between levels of socio-economic development and baseline area, which can be interpreted as the net attractive "force" an individual in that society exerted on others through their interactions. We shall see that patterns in the OECD data reinforce this idea.

## Contemporary data

To summarize patterns in the OECD data in a reasonably concise way we have aggregated FUA population and area estimates for various nations into regional groupings. Table 2 summarizes these groupings and the number of FUAs in each nation. Although the minimum population of an FUA in the OECD database is 50,000 persons, here we have removed FUAs with fewer than 100,000 persons to minimize the role of measurement error given the coarse precision of spatial measurement (1 $km^2$ blocks) in these data. Table 3 summarizes the resulting groupings and their population-area relationships; and Fig 3 presents scatterplots and fit lines, and quantile plots of the residuals, for these same groupings. Once again, in every case the relationship between log-transformed population and area is well-described by a linear function, with residuals that approximate a standard normal distribution.

Fig 4 provides a visual summary of the results, with cases listed in order of their respective scaling parameters. Fig 4A presents point estimates and confidence intervals for the scaling exponent $\beta$, once again showing the SST predicted range using dashed lines. Comparison of this plot with Fig 2A makes clear that the OECD data show much more variation than the archaeological and historical data. There are a number of systems from the developed world, including South Korea, Western Europe, North America, Japan, Northern Europe, and Southeastern Europe, where FUA areas become increasingly dense with population, and with exponents in the expected range of SST. In these systems, it would appear the OECD algorithm has defined spatial units that capture the infrastructural areas over which populations mix socially on a daily basis.

**Table 2. Regional groupings of the OECD data.**

| Group | Nation, FUAs (100,000+ inhabitants) |
|---|---|
| Australia | Australia, 18, New Zealand, 7 |
| Brazil | Brazil,182 |
| C America | Costa Rica, 2, El Salvador, 7, Guatemala, 20, Honduras, 8, Nicaragua, 9, Panama,4 |
| C Asia | Afghanistan, 22, Azerbaijan, 6, Georgia, 3, Kazakhstan, 20, Kyrgyzstan, 5, Mongolia, 1, Nepal, 5, Tajikistan, 5, Turkmenistan, 6, Uzbekistan, 32 |
| Caribbean | Bahamas, 1, Barbados, 1, Comoros, 2, Cuba, 15, Dominican Republic, 11, Haiti, 10, Jamaica, 4, Puerto Rico, 3, Saba, 1, Trinidad and Tobago, 3 |
| China | People's Republic of China, 1177, Chinese Taipei, 9, Hong Kong, 1, Macau, 1 |
| E Africa | Egypt, 51, Eritrea, 6, Ethiopia, 162, Kenya, 17, Madagascar, 2, Malawi, 3, Mauritius, 1, Mozambique, 27, Rwanda, 3, Somalia, 15, South Sudan, 13, Sudan, 54, Tanzania, 22, Zimbabwe, 9 |
| E Europe | Belarus, 12, Bulgaria, 6, Czech Republic, 12, Estonia, 2, Hungary, 9, Latvia, 1, Lithuania, 5, Moldova, 3, Poland, 35, Russia, 141, Slovak Republic, 6, Ukraine, 55 |
| India | India, 1193 |
| Japan | Japan, 84 |
| Mexico | Mexico, 114 |
| N Africa | Algeria, 51, Libya, 10, Mauritania, 2, Morocco, 34, Tunisia, 8, Western Sahara, 1 |
| N America | Canada, 32, United States, 250 |
| N Europe | Denmark, 4, Finland, 6, Iceland, 1, Norway, 4, Sweden, 11 |
| N Korea | Democratic People's Republic of Korea, 36 |
| Oceania | Fiji, 1, New Caledonia, 1, Papua New Guinea, 4, Togo, 8 |
| S Africa | Eswatini, 1, Lesotho, 1, South Africa, 37 |
| S America | Argentina, 38, Bolivia, 9, Chile, 22, Colombia, 47, Ecuador, 18, Guyana, 1, Paraguay, 6, Peru, 22, Suriname, 1, Uruguay, 3, Venezuela, 46 |
| S Asia | Bangladesh, 57, Jammu and Kashmir, 17, Pakistan, 128, Sri Lanka, 10 |
| S Korea | Korea, 27 |
| SE Asia | Cambodia, 3, Indonesia, 187, Lao People's Democratic Republic, 1, Malaysia, 26, Myanmar, 52, Philippines, 48, Singapore, 1, Thailand, 32, Timor-Leste, 1, Viet Nam, 63 |
| SE Europe | Albania, 4, Armenia, 2, Bosnia and Herzegovina, 4, Croatia, 4, Greece, 8, Kosovo, 3, Montenegro, 1, North Macedonia, 3, Romania, 24, Serbia, 5, Slovenia, 2, Turkey, 88 |
| W Africa | Angola, 31, Benin, 10, Botswana, 3, Brunei Darussalam, 1, Burkina Faso, 9, Burundi, 6, Côte d'Ivoire, 13, Cameroon, 29, Central African Republic, 1, Chad, 24, Congo, 6, Democratic Republic of the Congo, 74, Djibouti, 1, Equatorial Guinea, 2, Gabon, 1, Gambia, 2, Ghana, 22, Guinea, 8, Guinea-Bissau, 1, Liberia, 1, Mali, 7, Namibia, 1, Niger, 14, Nigeria, 215, Senegal, 17, Sierra Leone, 5, Uganda, 9, Zambia, 22 |
| W Asia | Bahrain, 1, Iran, 98, Iraq, 40, Israel, 9, Jordan, 5, Kuwait, 1, Lebanon, 6, Oman, 6, Qatar, 2, Saudi Arabia, 29, Syrian Arab Republic, 11, United Arab Emirates, 4, West Bank and Gaza Strip, 6, Yemen, 14 |
| W Europe | Austria, 6, Belgium, 10, Cyprus, 3, France, 70, Germany, 69, Ireland, 3, Italy, 61, Luxembourg, 1, Malta, 1, Netherlands, 25, Portugal, 7, Spain, 56, Switzerland, 13, United Kingdom, 90 |

There is a second set of systems where the scaling exponent is in the range $5/6 < \beta < 1$. These systems, which include Mexico, Brazil, Australia, Eastern Europe, China, Central Asia, South America, and South Africa, still exhibit densification with population, but at a more modest rate than is predicted by SST. At least in some cases, these estimates are consistent across analyses using different types of spatial units. For example, Zünd and Bettencourt [61] estimated $\beta = .88$ for the relationship between population and total area of prefectural cities in China in 2014, and Jiao and others [62] estimated $\beta = .87$ for the relationship between population and built-up area for prefectural cities in 2016. Both are indistinguishable from the exponent for the FUA population-area relationship in the OECD dataset. Several of these regions have experienced rapid growth in recent decades, with significant state investment, and it is tempting to

**Table 3. Population-area relationships in OECD groupings.**

| Region | Sample Size | Slope | SE | Intercept | SE | $R^2$ |
|---|---|---|---|---|---|---|
| Australia | 25 | .86 | .084 | .37 | 1.085 | .82 |
| Brazil | 182 | .85 | .034 | -.71 | .430 | .77 |
| C America | 50 | 1.00 | .096 | -2.72 | 1.186 | .69 |
| C Asia | 105 | .92 | .067 | -2.15 | .835 | .65 |
| Caribbean | 51 | .99 | .118 | -3.03 | 1.457 | .59 |
| China | 1188 | .88 | .012 | -1.29 | .148 | .83 |
| E Africa | 385 | 1.38 | .058 | -9.23 | .715 | .60 |
| E Europe | 287 | .88 | .036 | -.55 | .450 | .68 |
| India | 1193 | 1.10 | .026 | -5.36 | .323 | .61 |
| Japan | 84 | .79 | .019 | .65 | .244 | .96 |
| Mexico | 114 | .84 | .046 | -.53 | .585 | .75 |
| N Africa | 106 | 1.04 | .053 | -3.77 | .656 | .79 |
| N America | 282 | .79 | .026 | 1.63 | .331 | .77 |
| N Europe | 26 | .80 | .076 | 1.24 | .967 | .82 |
| N Korea | 36 | 1.21 | .123 | -6.52 | 1.503 | .74 |
| Oceania* | 14 | 1.27 | .315 | -7.31 | 3.838 | .58 |
| S Africa | 39 | .97 | .054 | -2.59 | .681 | .90 |
| S America | 213 | .94 | .043 | -2.33 | .549 | .69 |
| S Asia | 212 | 1.03 | .046 | -3.83 | .587 | .70 |
| S Korea | 27 | .71 | .029 | 1.44 | .385 | .96 |
| SE Asia | 414 | 1.02 | .029 | -3.36 | .365 | .75 |
| SE Europe | 148 | .83 | .058 | -.47 | .727 | .58 |
| W Africa | 535 | 1.13 | .039 | -5.57 | .482 | .61 |
| W Asia | 232 | .99 | .036 | -3.19 | .458 | .77 |
| W Europe | 415 | .78 | .026 | 1.10 | .331 | .68 |

Note: Unless otherwise noted, regressions are significant at the P < .0001 level.

*P = .00161

speculate that faster rates of land development are responsible for these slightly larger exponents. The expectation of SST for a networked city is based on the idea that infrastructure is added incrementally as the city grows, such that the amount that is added is proportional to the population density at each time step. If growth is rapid, infrastructural area may be added more slowly or more rapidly than will be needed for the city at that time. Notice that if this were true across the system it would also affect the pre-factor or baseline area for that system.

Finally, there are several regions where the scaling exponent is in the range $1<\beta<1.5$. These systems are concentrated in the developing world, from the Caribbean to the Middle East, Central America, Southern Asia, Africa, and Oceania. In these systems there are good reasons to believe OECD FUAs over-estimate the areas over which the social interactions of the enclosed groups occur. To give just one example, Sahasranaman and Bettencourt [63] estimated $\beta = .88$ for urban agglomerations defined by the 2011 Census of India. An urban agglomeration in that census is a continuous urban spread constituting a town and its adjoining "urban outgrowths", or two or more physically contiguous towns and adjoining urban outgrowths of such towns. An urban outgrowth is in turn defined as a viable unit, contiguous to a statutory town, possessing urban features in terms of infrastructure and amenities. These units seem to draw closer boundaries around the infrastructural area than the OECD data, and the estimated scaling exponent is more in keeping with SST.

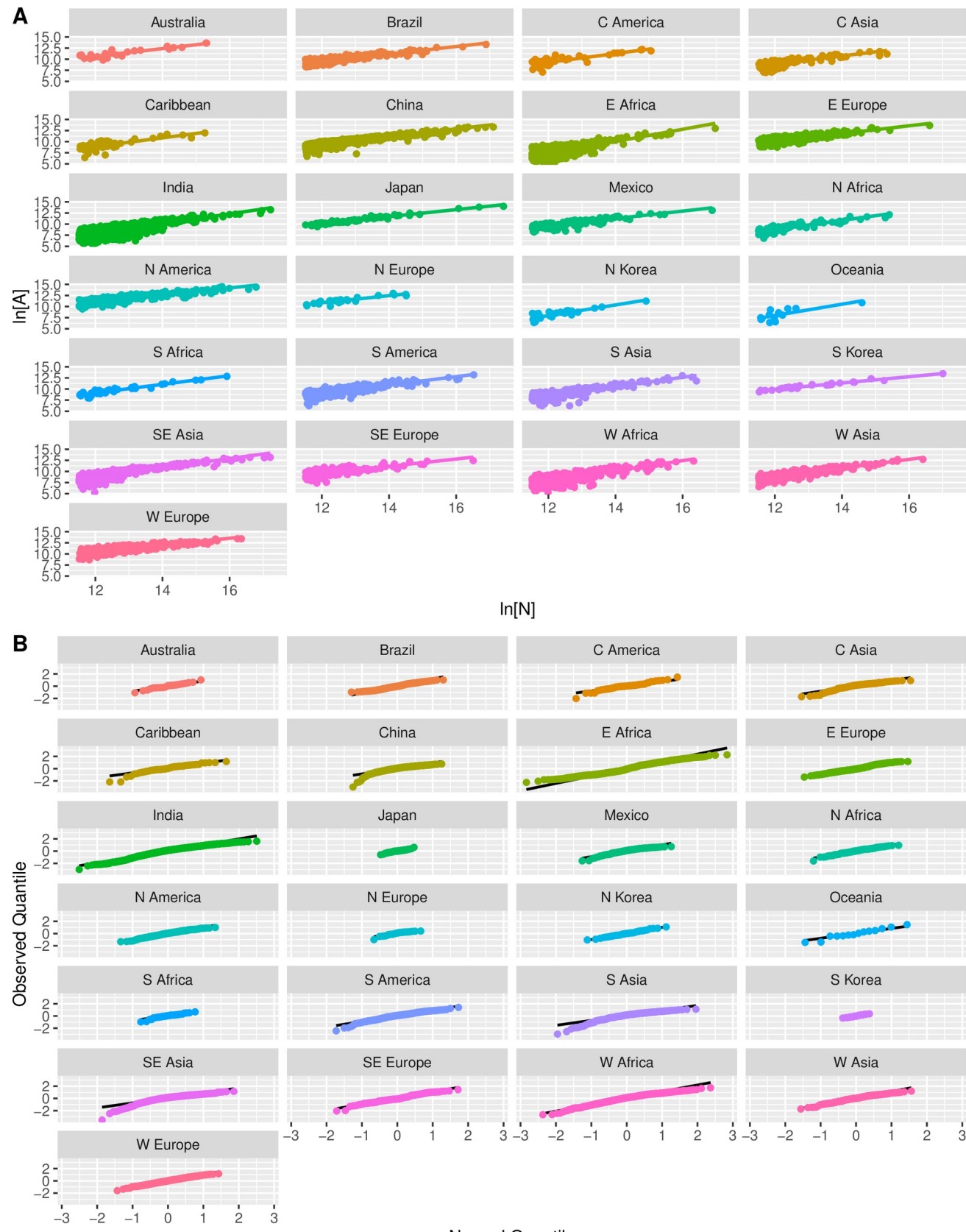

**Fig 3.** Population-area relationships for OECD groupings: A) regression plots; B) quantile plots of residuals.

We can think of at least two possible reasons why the OECD data for the developing world do not correspond to SST predictions. One possibility is that when applied to the developing world the OECD algorithm defines areas that contain increasing fractions of rural or non-urban land. For example, if urban growth follows a radial pattern, along roads leading to/from existing centers, workers who live along these roads may commute into the urban area for work, but the areas where they live will not be as interconnected as the commuting zones (sub-urban areas) of FUAs in developed nations. As a result, larger FUAs will contain higher fractions of unbuilt area, as occurs in a few of the archaeological cases discussed earlier. Note that we are not suggesting that the residents of cities in the developing world do not take advantage of the economies of scale that characterize cities in developed nations. Instead, we are suggesting that in the developing world the areas over which urban populations mix are sparser, more intricate, and less fully connected spatially, such that these areas are not readily captured by contiguous local administrative or census units that are aggregated through the OECD algorithm.

A second possibility for why the OECD data do not conform to SST predictions for the developing world is that one or more parameters of the pre-factor of the scaling relation are not scale independent. If, for example, transport costs are less in larger cities than in smaller ones this would add to the exponent of the overall scaling relation. To see this, we can first re-write Eq (3) above as $A_n = (\hat{g} a_0 l/\epsilon)^{1/3} N^{5/6}$. If transport costs are lower in larger cities due to the use of different transport technologies, such that $\epsilon(N) = \epsilon_0 N^{-\gamma}$, then $A_n = (\hat{g} a_0 l/\epsilon_0)^{1/3} N^{5/6+\gamma/3}$. In this situation, any value of $\gamma > 1/2$ will lead to area increasing faster than population. It is beyond the scope of this paper to investigate either of these possibilities further, but we raise them to illustrate how theory is helpful even when one obtains unexpected or inconsistent results. In such cases, either the theory is right, but the data do not reflect the parameters of the theory; or the theory is missing something and needs to be adjusted to take the new results into account. In other words, having concrete expectations is an important aid to continuing progress in understanding urban phenomena. In the absence of such expectations, it is much harder for knowledge to accumulate.

Fig 4B summarizes the point estimate and confidence interval for the pre-factor of the population-area relationship across our OECD groupings. As argued above, the baseline area reflects the net attractive force on others exerted by an individual through their interactions, and thus represents an index of an important dimension of socio-economic development. Time series analysis of the population-area relationship reinforces this view. Fig 5, for example, shows the population-area relationship for each five-year period between 1960 and 2005 for major metropolitan areas in Japan (Data obtained at https://www.stat.go.jp/english/data/index.html). The plot shows that the slope of the fit line is consistent over time and with SST expectations (the average slope across time steps is $\hat{\beta} = .858 \pm .023$), but the intercept of the fit line increases over time from $a_{1960} = .128 \mathrm{m}^2$ to $a_{2005} = 3.274 \mathrm{m}^2$. Notice also that there was relatively little change in the intercept between 1990 and 2005, a period generally recognized as one of stagnation in the Japanese economy. Such results indicate that an important dimension of socio-economic development is increases in the net attractive force of an individual's movements and interactions, reflected in the magnitude of the baseline area.

Two patterns in the OECD results reinforce this interpretation. First, the confidence intervals for these estimates are much smaller, primarily due to more consistent and temporally precise data than is the case for the archaeological and historical cases. Second, the smallest

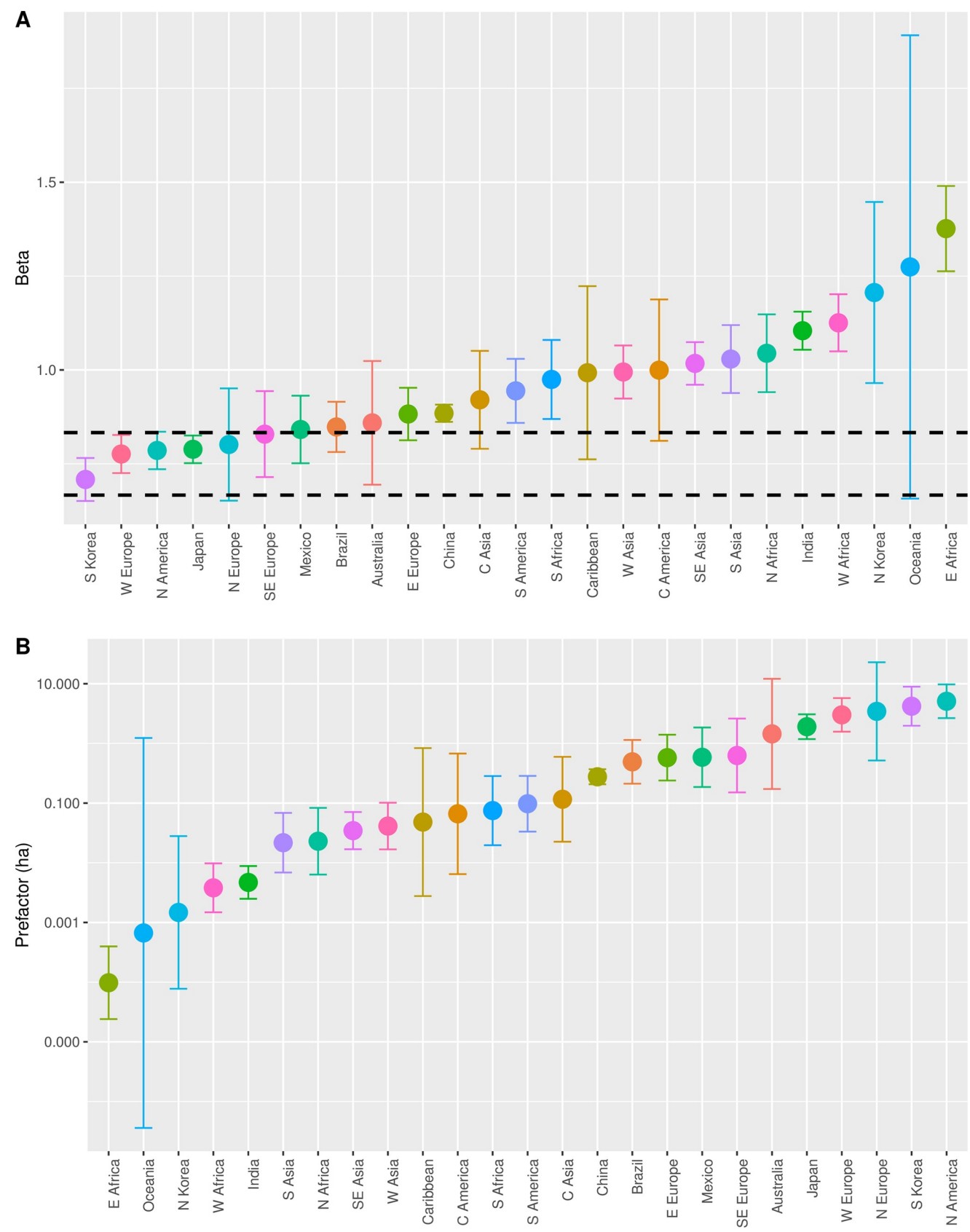

**Fig 4.** Scaling relations for OECD groupings: A) exponents; B) pre-factors.

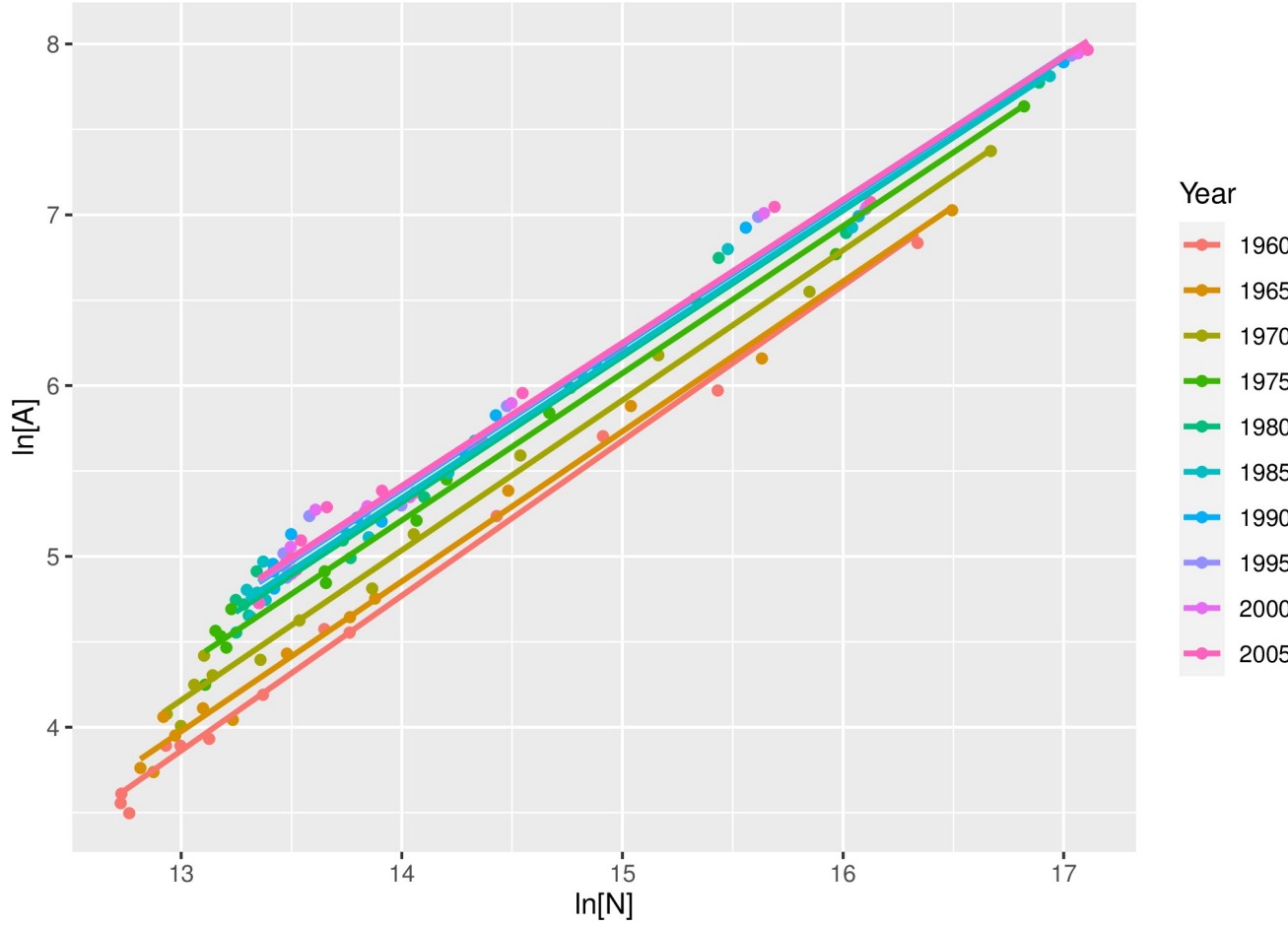

**Fig 5. Population-area relationship for major metropolitan areas in Japan, 1960–2005.**

baseline areas in the OECD groupings, from East Africa, Oceania, North Korea, West Africa, and India, are of comparable magnitude to the baseline areas of the archaeological and historical systems of the preindustrial world. This suggests the net attractive force generated by an individual's movements and interactions in the least-developed regions of the world today are not appreciably different than they were in preindustrial societies of the past. These patterns suggest baseline areas derived from the population-area relationship track an important dimension of socio-economic development and that one can interpret the variation in Fig 4B in the same terms. Given this, Fig 4B indicates, for example, that the attractive force of an individual's movements and interactions is much higher in South Korea than it is in North Korea. Notice also that due to the elasticity of the scaling relationship these baseline areas do not reflect the area per person across all cities, or even the average area per person across cities. Rather, they reflect the area an individual can take up in the smallest settlements given the degree to which others are attracted to them.

We can deepen this interpretation by considering the relationship between exponents and pre-factors. Fig 6 presents a scatterplot of the relationship between exponents and pre-factors across our OECD groupings. The negative correlation between the exponent and pre-factor across regions is striking and suggests that the population-area relationship for FUAs across urban systems reveals much about global patterns of urban development. When the exponent

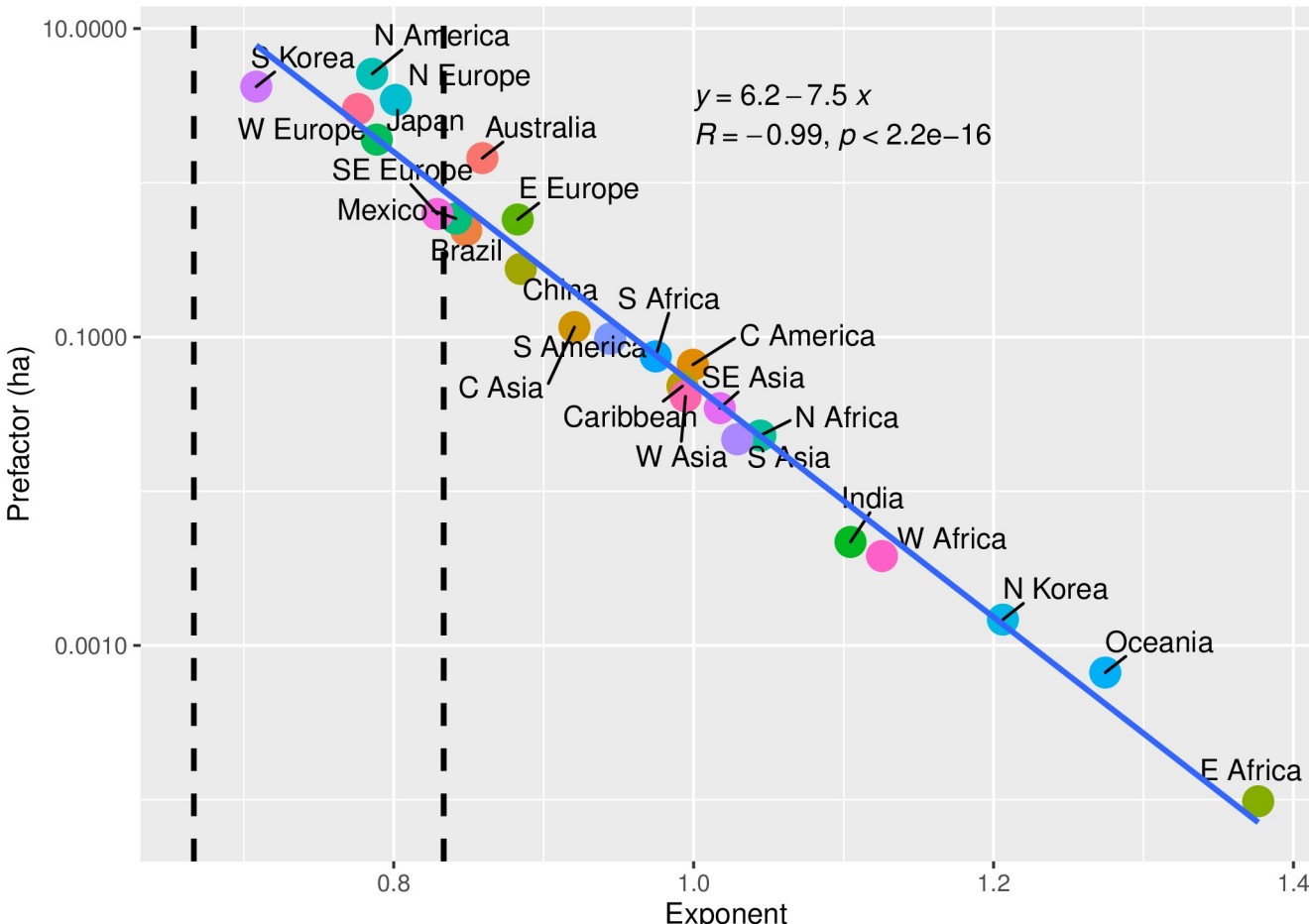

**Fig 6. Relationship between exponents and pre-factors for OECD groupings.**

of this relationship is less than one the total area per person declines with city size, and when it is greater than one this area increases with city size. However, due to the issues with FUA definitions we have discussed, we suspect the rate of change in infrastructural area per person is more consistent across systems than these data suggest. Keeping in mind the possibility that FUAs do not consistently measure infrastructural area across levels of socio-economic development, what these data suggest is that individuals currently generate weaker attractive forces in the developing world; that variation in *geographical* area per person (though not necessarily the *network* area per person) is much more extreme between large and small cities in the developing world; and that increases in baseline area are connected to spatial patterns of development which lead to variation in the relationship between population and geographical area. In general, larger baseline areas correlate with decentralized urban infrastructure networks, reflected in scaling exponents in the range defined by SST; and smaller baseline areas seem to correlate with more centralized networks which lead to the inclusion of rural land within FUAs, reflected in scaling exponents that are larger than predicted by SST.

## Conclusions

In this paper we have discussed theoretical definitions of cities and settlements, various means of defining and measuring their boundaries, and a theoretical approach that makes specific

quantitative predictions regarding the average relationship between population and area across settlements in a system. We also highlighted the ways in which this approach incorporates key ideas from the study of complex systems. We then examined data for a range of archaeological and historical systems, and the OECD global human settlement database, to assess the degree to which these systems, when measured in specific ways, correspond to these predictions. The fact that SST predictions hold across many systems—ranging over thousands of years, multiple continents, several orders of magnitude in scale, substantial variation in social and political institutions, and radically-different levels of technological and socio-economic development— suggests that SST is on the right track with respect to capturing fundamental social processes that drive agglomeration effects in cities everywhere. We have also observed variation in the pre-factors of scaling relationships across systems which is consistent with a general interpretation of this number as an index of the net attractive force exerted by an individual through their movements and interactions. This index represents an important dimension of socio-economic development that is scale-independent, as can be seen by the overlapping values observed for ancient settlement systems and the least developed regions of the world today, and represents an important avenue for additional research on the determinants of socio-economic development.

We hope the analyses presented here will persuade readers that, although the inner workings of cities are difficult to quantify in detail, and even aggregate properties can be difficult to measure, it is productive to work with simple models of their internal dynamics that make predictions regarding relationships among the properties of cities across the world and over time. These simple models offer two advantages over a more inductive approach. First, to the extent that they account for certain regularities, they help focus investigations of the dimensions and determinants of variation across systems after taking the regularity into account. Second, when the data do not conform to expectations of concrete simple models, the existence of such models help one to ask sharper questions regarding why. All of this is possible precisely because cities are complex systems, not despite it.

As noted above, in several cases the observed relationships between population and area do not correspond with expectations of SST. One possible explanation for these cases has to do with the degree to which social mixing took place over the areas of spatial units; and another possibility is that the theory requires adjustment. Nevertheless, such unexpected results do not undermine the prospect of a predictive theory of cities and urbanization; rather, they point out that empirical work on cities must be in dialog with theory. In this case, asking why population-area relationships do not conform to SST in specific circumstances suggests productive avenues for further research and leads to several potential insights on urban development. Following such leads may eventually lead to a conclusion that current models in SST are incorrect. But in doing so, better models will be developed to replace them, and actionable knowledge of the fundamental nature of cities will have improved. In other words, the true advantage of having a concrete predictive theory to guide research is not merely the ability to test ideas and see if the predictions hold. Rather, it is the ability to ask why the prediction does not hold in certain situations and to pursue potential answers to those questions. We think this is especially important for a nascent field like urban science where, in many ways, practitioners are just figuring out the scope and purposes of the field. We believe this spirit of investigation is far more important than the specifics of any particular theory, model or result. We hope this exploration illustrates the value of predictive theory, not so much as a final destination but as a flashlight that helps one determine where to walk in the darkness. Although empirical challenges will always remain, predictive theory—inspired and informed by robust insights garnered by complexity science—is both possible and helpful for the larger goals of urban science.

## Acknowledgments

We thank Steven Kowalewski for providing access to archaeological survey data from the Mixteca Alta region.

## Author Contributions

**Conceptualization:** Scott G. Ortman, José Lobo, Michael E. Smith.

**Data curation:** Scott G. Ortman.

**Formal analysis:** Scott G. Ortman.

**Funding acquisition:** Scott G. Ortman.

**Investigation:** Scott G. Ortman, José Lobo, Michael E. Smith.

**Methodology:** Scott G. Ortman, José Lobo.

**Project administration:** Scott G. Ortman.

**Visualization:** Scott G. Ortman.

**Writing – original draft:** Scott G. Ortman, José Lobo, Michael E. Smith.

**Writing – review & editing:** Scott G. Ortman, José Lobo, Michael E. Smith.

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
