## [Decision Letter · Decision Letter 0]

6 Oct 2020

PONE-D-20-28450

Cities: Complexity, Theory and History

PLOS ONE

Dear Dr. Ortman,

Thank you for submitting your manuscript to PLOS ONE. After careful consideration, we feel that it has merit but does not fully meet PLOS ONE’s publication criteria as it currently stands. Therefore, we invite you to submit a revised version of the manuscript that addresses the points raised during the review process.

All comments need to be addressed before re-submission.

We look forward to receiving your revised manuscript.

Kind regards,

Peter F. Biehl, PhD

Academic Editor

PLOS ONE

Journal Requirements:

"Portions of this research were supported by a grant from the James S. McDonnell Foundation (#220020438, to SO): https://www.jsmf.org/. ".

i) Please provide an amended statement that declares *all* the funding or sources of support (whether external or internal to your organization) received during this study, as detailed online in our guide for authors at http://journals.plos.org/plosone/s/submit-now.  Please also include the statement “There was no additional external funding received for this study.” in your updated Funding Statement.

ii) Please include your amended Funding Statement within your cover letter. We will change the online submission form on your behalf.

Additional Editor Comments (if provided):

Your manuscript has now been seen by three referees, whose comments are appended below. You will see from these comments that the referees find your work of great interest, but have raised several concerns that must be addressed before re-submission.

Reviewers' comments:

Reviewer's Responses to Questions

**Comments to the Author**

1. Is the manuscript technically sound, and do the data support the conclusions?

Reviewer #1: Yes

Reviewer #2: Yes

Reviewer #3: Yes

2. Has the statistical analysis been performed appropriately and rigorously? 

Reviewer #1: Yes

Reviewer #2: Yes

Reviewer #3: Yes

3. Have the authors made all data underlying the findings in their manuscript fully available?

Reviewer #1: Yes

Reviewer #2: No

Reviewer #3: Yes

4. Is the manuscript presented in an intelligible fashion and written in standard English?

Reviewer #1: Yes

Reviewer #2: Yes

Reviewer #3: Yes

5. Review Comments to the Author

Reviewer #1: This is an interesting paper which shows that there is sufficient correlation to suggest that the relationship between population size and settlement area can be predicted b Settlement Scaling Theory. That is, that cities are complex systems whose basic features (area and population size) nevertheless follow a relatively simple rule. The analysis reveals that there are cities whose area is not predicted by their population (or vice versa) but suggests that explanations can be found for these anomalies and therefore that the anomalous cases confirm the model. That seems plausible, although there is sufficient uncertainty in the data that the levels of overall confidence must be limited, particular with regard to archaeological cases where, as the authors acknowledge, in many cases evidence for population and evidence for area are not independent but scholars have derived population from area or from some aspect of settlement that directly relates to area.

The major question for the paper is whether it needs the long general discussion that is front-loaded, in which 'what is a city?' and why theory might be useful are discussed. For my own part, I thought that these could be dispensed with and the paper could start more or less at p. 9.

The importance of the paper lies partly in the anomalous cities and the explanations for their anomaly, partly in the emphasis on the independence of the size: population relationship from any particular type of economy, and partly on the insistence that the relationship applies to settlements of more or less every size.

Reviewer #2: The analysis presented by Ortman, Lobo, and Smith of archeological, historic, and contemporary settlement data demonstrates very effectively the consistent impact of distance-interaction principles on the internal residential density of settlements across a wide range of population sizes, time periods, and technological regimes. The authors do not describe what they do in these terms, and some readers might find an easier path into the abstract mathematical treatment of the subject if they did. The fundamental principle the article revolves around is that settlements with larger populations tend to be more densely packed than settlements with smaller populations. As a general principle, this jibes perfectly well with common sense and traditional wisdom. Ortman, Lobo, and Smith carry the idea farther with a crisp mathematical characterization of the relationship between population size and spatial size (which is to say, density), based on a formalization of several fundamental assumptions that will seem inherently plausible to most readers who have thought much about the dynamics of human settlements.

This mathematical characterization of the relationship enables them to place in the same comparative frame a breathtaking array of settlement systems averaged across long time spans for single regions for the archeological cases and across broad swaths of the globe for modern times. One might well complain about such broad averaging together of up to 8,000 years of settlement change or of contemporary cities as varied as those of an entire continent, such as South America, or substantial part of one, such as West Africa. But this is one effective approach to establishing a comparative baseline of very broadly generalized consistent patterns. Just as Ortman, Lobo, and Smith observe, it is worthwhile and exciting to establish such a baseline with what might seem mathematical overkill primarily because it sharpens the ability to identify cases that do not fit the general pattern. Investigation of the factors underlying these departures holds the most promise for unexpected enlightenment about settlement dynamics that go beyond the fundamental tendency for settlements with larger populations to be more densely packed.

For Ortman, Lobo, and Smith, it is the somewhat cryptically named "pre-factor" that puts a number on the circumstances that vary across space and time and can cause settlements with larger populations to be less densely packed or even more densely packed than expected on the basis of the average tendency. This number does point clearly to such "deviant" settlement systems and identify the direction and strength of their differences from the "normal" relationship between population size and residential density, but because the general relationship is established in terms of such broad averaging, much potentially important variation is subsumed within the averages (and thus lost sight of), and the "pre-factor," in itself, does little to help us actually understand variation.

Ortman, Lobo, and Smith do offer a few more specific hints about directions that exploration of this variation could take. One in particular is connected to both methodological and conceptual challenges in research on this subject. Classic Maya and Izapa settlement systems and dispersed settlements in the Basin of Mexico are singled out among the archeological cases here because settlements with larger populations tend to be less densely settled than those with smaller populations. The same occurs for several parts of the contemporary world: Africa, Oceania, South and Southeast Asia. Ortman, Lobo, and Smith suggest that this is essentially a methodological issue of figuring out just how to draw a proper line around a settlement in space. And there is, indeed, a methodological issue here, leading to concepts like Metropolitan Statistical Areas or Functional Urban Areas for contemporary cities, and arbitrary rules setting 100 m as the size of gaps in surface artifact distributions that indicate separate archaeological sites which are then automatically assumed to be separate social communities.

Drennan, Berrey, and Peterson (2015) are cited as justification for this archeological rule, although actually it is argued there (pp. 52-61) that this is an arbitrary and misleading approach that can split socially meaningful settlements into parts or even create "settlements" where none exist. The fact that it is very hard to draw a line around settlements in some systems is not just a methodological challenge to get around; it is a message about substance. The cases enumerated in the previous paragraph are ones in which the social phenomenon of "settlement" and the interaction that constitutes one are truly different in character—more diffuse and less bounded in reality than in cases where the boundaries of settlements are pretty clear. In extreme cases there may be no local community structure at all, just scattered farmsteads. Insisting on seeing all settlement systems in terms of clearly bounded settlement units that correspond to bounded local interaction communities blinds us to important aspects of interaction structure. And, well, this is but one of the directions that exploring deviations from Ortman, Lobo, and Smith's general averages can productively go.

Ortman, Lobo, and Smith's skillful analysis of previously collected data is thought-provoking, and, one hopes, will encourage other such efforts to make existing information tell us more. Firm connection to the data the analysis is based on is essential, and here there is room for improvement. I'll confine myself to the archeological datasets, since that's where my expertise lies. The complete supporting data are not actually available in the tDAR repository cited. No data are there on the Lower Santa Valley, the Mixteca Alta, Chifeng, the Classic Maya, or Izapa. For some of these regions the original published sources are cited, but this leaves a lot unaccounted for in how those published sources were translated into the datasets analyzed for this article. A large spreadsheet provides Basin of Mexico data, settlement by settlement, and Ortman et al. (2014) discuss how that spreadsheet became the analysis in the 2014 article, but the analysis in this article is different. The sort of accounting the 2014 article provides is needed here too. As yet unpublished articles might clarify the Chifeng, Classic Maya, and Izapa datasets, but as of now the data provided on line do not make it possible to track all the archeological analyses back to the primary data.

Reviewer #3: This is a very interesting paper which demonstrates the potential of applying Settlement Scaling Theory (SST) to improve our understanding of cities in the past and present. The paper is well-written and structured, and it incorporates a large amount of data from different regions around the world. The methodology applied in the paper and its ambitious scope assures a wide reception and notable impact not only in archaeology, but also in a variety of other disciplines. I recommend publication without any hesitation.

As a minor recommendation, I think the paper would benefit from less direct quotes; some of the existing ones could easily be paraphrased and/or shortened.

The second part of the bibliography is confusing, why are the references in the section "References added by Jose" not incorporated into the general bibliography?

6. PLOS authors have the option to publish the peer review history of their article (what does this mean?). If published, this will include your full peer review and any attached files.

Reviewer #1: No

Reviewer #2: **Yes: **Robert D. Drennan

Reviewer #3: **Yes: **Dr Manuel Fernández-Götz

---

## [Author Response · Author response to Decision Letter 0]

4 Nov 2020

11/2/2020

Dear Editors:

Thank you for soliciting three helpful reviews of our article. Before proceeding to address the reviewers' comments we want to re-emphasize that our intention was to submit a manuscript for consideration of the special issue on cities and complexity. We communicated our intention to do so to Marta Gonzalez and Diego Rybski, the two guest editors of the planned special issue. In writing the submitted paper we intended the audience to be general urban scientists and not necessarily archaeologists. So although we are pleased that the archaeologists who reviewed our paper found that the data and analysis supported our conclusions, we would appreciate it if the submission was considered for the special issue and if it were to be shared with the two guest editors.

Reviewer #1: 

The major question for the paper is whether it needs the long general discussion that is front-loaded, in which 'what is a city?' and why theory might be useful are discussed. For my own part, I thought that these could be dispensed with and the paper could start more or less at p. 9.

• We thank the reviewer for pointing out that the initial sections of our paper are longer than they should be. However, we do not think cutting these passages entirely is appropriate because we wrote our paper for the special collection “cities and complexity” and hope it will be read by urbanists interested in complexity, and complexity scientists interested in cities. Still, we take the reviewer’s comment as an indication that we could do more to connect these general considerations to the analysis and results. So we have restructured and shortened the initial sections of the paper in an effort to accomplish this.

Reviewer #2: 

The analysis presented by Ortman, Lobo, and Smith of archeological, historic, and contemporary settlement data demonstrates very effectively the consistent impact of distance-interaction principles on the internal residential density of settlements across a wide range of population sizes, time periods, and technological regimes. The authors do not describe what they do in these terms, and some readers might find an easier path into the abstract mathematical treatment of the subject if they did. 

• We are very pleased that the reviewer is convinced by our analyses of the archaeological and OECD data, and we thank this person for pointing out that distance-interaction principles are an additional way in which one could introduce our approach. Since the paper is intended primarily for the community of researchers interested in cities and complexity, we think the existing approach is more appropriate for this paper. But we will certainly keep this idea in mind for future efforts geared toward different audiences. 

For Ortman, Lobo, and Smith, it is the somewhat cryptically named "pre-factor" that puts a number on the circumstances that vary across space and time and can cause settlements with larger populations to be less densely packed or even more densely packed than expected on the basis of the average tendency. 

• We thank the reviewer for pointing out that our terminology may appear cryptic to some readers, and we have added a few comments in the relevant places in an effort to clarify the meaning of the term “pre-factor”. 

This number does point clearly to such "deviant" settlement systems and identify the direction and strength of their differences from the "normal" relationship between population size and residential density, but because the general relationship is established in terms of such broad averaging, much potentially important variation is subsumed within the averages (and thus lost sight of), and the "pre-factor," in itself, does little to help us actually understand variation.

• We agree that the pre-factor of a scaling relation is a measure of a system-level property that does not contribute to the understanding of variation across individual settlements within the system. However, the pre-factor does help one identify which sorts of variation are significant, given the properties of the system. We have added language to the discussion to emphasize this further. 

And there is, indeed, a methodological issue here, leading to concepts like Metropolitan Statistical Areas or Functional Urban Areas for contemporary cities, and arbitrary rules setting 100 m as the size of gaps in surface artifact distributions that indicate separate archaeological sites which are then automatically assumed to be separate social communities.

Drennan, Berrey, and Peterson (2015) are cited as justification for this archeological rule, although actually it is argued there (pp. 52-61) that this is an arbitrary and misleading approach that can split socially meaningful settlements into parts or even create "settlements" where none exist. 

The fact that it is very hard to draw a line around settlements in some systems is not just a methodological challenge to get around; it is a message about substance. The cases enumerated in the previous paragraph are ones in which the social phenomenon of "settlement" and the interaction that constitutes one are truly different in character—more diffuse and less bounded in reality than in cases where the boundaries of settlements are pretty clear. In extreme cases there may be no local community structure at all, just scattered farmsteads. Insisting on seeing all settlement systems in terms of clearly bounded settlement units that correspond to bounded local interaction communities blinds us to important aspects of interaction structure. And, well, this is but one of the directions that exploring deviations from Ortman, Lobo, and Smith's general averages can productively go.

• This is an excellent point and we have inserted some discussion to this effect in the paper. 

The complete supporting data are not actually available in the tDAR repository cited. 

• We were waiting to post the data until our paper was closer to acceptance, but we have done so now. 

Reviewer #3: 

As a minor recommendation, I think the paper would benefit from less direct quotes; some of the existing ones could easily be paraphrased and/or shortened.

• We have removed some of the direct quotes as part of re-structuring the initial part of the paper, but we feel the ones that remain are important because they provide strong justification for our approach.

The second part of the bibliography is confusing, why are the references in the section "References added by Jose" not incorporated into the general bibliography?

• We thank the reviewer for pointing out this formatting issue. We have fixed it in the revised version.

---

## [Decision Letter · Decision Letter 1]

25 Nov 2020

Cities: Complexity, Theory and History

PONE-D-20-28450R1

Dear Dr. Ortman,

We’re pleased to inform you that your manuscript has been judged scientifically suitable for publication and will be formally accepted for publication once it meets all outstanding technical requirements.

Kind regards,

Peter F. Biehl, PhD

Academic Editor

PLOS ONE

Additional Editor Comments (optional):

Reviewers' comments:

Reviewer's Responses to Questions

**Comments to the Author**

1. If the authors have adequately addressed your comments raised in a previous round of review and you feel that this manuscript is now acceptable for publication, you may indicate that here to bypass the “Comments to the Author” section, enter your conflict of interest statement in the “Confidential to Editor” section, and submit your "Accept" recommendation.

Reviewer #2: All comments have been addressed

2. Is the manuscript technically sound, and do the data support the conclusions?

Reviewer #2: Yes

3. Has the statistical analysis been performed appropriately and rigorously? 

Reviewer #2: Yes

4. Have the authors made all data underlying the findings in their manuscript fully available?

Reviewer #2: Yes

5. Is the manuscript presented in an intelligible fashion and written in standard English?

Reviewer #2: Yes

6. Review Comments to the Author

Reviewer #2: (No Response)

7. PLOS authors have the option to publish the peer review history of their article (what does this mean?). If published, this will include your full peer review and any attached files.

Reviewer #2: **Yes: **Robert D. Drennan

---

## [Editor Report · Acceptance letter]

27 Nov 2020

PONE-D-20-28450R1 

Cities: Complexity, Theory and History 

Dear Dr. Ortman:

I'm pleased to inform you that your manuscript has been deemed suitable for publication in PLOS ONE. Congratulations! Your manuscript is now with our production department. 

Kind regards, 

on behalf of

Dr. Peter F. Biehl 

Academic Editor

PLOS ONE